# The Impact of Financial Development on Economic Growth in Nigeria: An ARDL Analysis

**Eugene Iheanacho**

Department of Economics, Abia State University, P.M.B. 2000, Uturu, Nigeria; dreugeneiheanacho@gmail.com; Tel.: +234-810-731-2206

**Abstract:** This study empirically examines the relationship between financial intermediary development and economic growth in Nigeria over the period 1981–2011 using the auto-regressive distributed lag (ARDL) approach to co-integration analysis. The results show that the relationship between financial development and economic growth in Nigeria is not significantly different from what has been observed generally in oil-dependent economies. The relationship between financial intermediary development and economic growth in Nigeria is found to be insignificantly negative in the long-run and significantly negative in the short-run. The results highlight the dominant role of the oil sector in economic activities in Nigeria.

**Keywords:** financial development; economic growth; oil price; oil-dependent economies; Nigeria; ARDL

## 1. Introduction

Theoretical explanations suggest that financial sector intermediary development stimulates economic growth by creating economic conditions that enhance efficiency in resource allocation (see Levine, 2004 [1]). Building on this theoretical foundation, a number of empirical studies examined the relationship between financial sector development and economic growth (see Chang and Caudill, 2005 [2]; Seetanah, 2008 [3]; Anwar and Nguyen, 2011 [4]; Uddin et al., 2013 [5]; Nwani and Bassey Orie, 2016 [6] among others). The results of most of these studies show that financial intermediary development is a significant driver of economic growth. Three major components of the financial intermediary system have been widely considered in these studies: the role of financial intermediaries in the mobilization of savings, the role of financial intermediaries in enhancing economic activities in the private sector and the size of the financial intermediary system. According Levine (1997 [7]), Levine (2004 [1]) and Beck et al. (2011 [8]) the growth-generating ability of every financial intermediary system depends significantly on how efficient the system could mobilize and allocate savings in the economy. By attracting deposits from various economic units in the economy and financing investment projects in the private sector, financial intermediaries generate higher levels of economic growth, support firms that depend on external finance and reduce the financing constraints of small- and medium-sized enterprises (Beck et al., 2005 [9]; Beck and Demirguc-Kunt, 2006 [10]; Beck et al., 2011 [8]).

However, results of recent time series studies on financial sector development and economic growth nexus in oil-exporting countries suggest weak or even negative relationship (see Cevik and Rahmati, 2013 [11]; Quixina and Almeida, 2014 [12]; Samargandi et al., 2014 [13]; Adeniyi et al., 2015 [14]; Nwani and Bassey Orie, 2016 [15] among others). The argument has been that most oil-exporting countries depend significantly on oil revenue and are unable to develop competitive financial sector that could stimulate economic activities in the private sector. Using a sample of

oil-exporting and non-oil exporting countries, Nili and Rastad (2007 [15]) show that financial sector development has less influence in oil-exporting economies due to the general inefficiency of financial institutions, the dominant role of public sector in resource allocation and the weakness of private sector in oil-exporting economies. Beck (2011 [16]) concludes that oil-dependent countries may be subject to natural resource curse in financial sector development. Examining the degree of reliance on natural resources, the study shows that countries that depend significantly on exports of oil tend to have underdeveloped financial systems and offer less credit to the private sector, despite the fact that banks in these countries are highly profitable and better capitalized. Barajas et al. (2013 [17]) used a dataset consisting of more than 130 countries from 1975 to 2005 to examine the finance-growth nexus by income level, across regions and between oil-exporting and non-oil exporting countries. In the particular case of oil-exporting countries, the study found the growth generating ability of financial intermediary development to be weak. The results suggest that the influence of banking sector development on economic growth in oil-exporting countries become weaker as the degree of oil dependence increase.

This study seeks to contribute to the existing studies by examining the case of Nigeria using the auto-regressive distributed lag (ARDL) approach to cointegration analysis. The Nigerian financial sector has witnessed a number of policy reforms over the past two decades, expected to stimulate the growth of the economy through increased mobilization of savings, provision of credit to the private sector and reduction in information and transaction costs making this study necessary. This study contributes to the few existing studies in two areas. First, by employing three financial development indices constructed using principal component analysis (PCA) to examine the sensitivity of finance-led-growth analysis to alternative combinations of individual indicators of financial development in Nigeria. Second, by including crude oil price among the control variables this study controls for the possible influence of oil on economic activities in Nigerian economy. The results of this study would be of interest to researchers and policy makers seeking to understand the role of financial sector development in Nigeria and other developing oil-exporting countries.

The structure of this article is as follows: Section 2 presents some stylized facts on financial intermediary development in Nigeria. Section 3 describes the data and empirical methodology used. Section 4 presents and discusses the empirical results. Finally, Section 5 concludes and offers some policy implications.

## 2. Some Stylized Facts

Table 1 presents a general overview of the performance of the Nigerian economy in comparison to the Algerian and a group of non-oil dependent economies (see Appendix A) using average per capita income and four financial intermediary development indicators over the period 1981–2011.

**Table 1.** Comparative analysis: GDP per capita and financial intermediary indicators in Nigeria, Algeria and some selected non-oil dependent countries.

|  | OPEC Member Country | | Some Non-Oil Dependent Countries |
|---|---|---|---|
|  | **Nigeria** | **Algeria** |  |
| GDP per capita (constant 2005 US$) | 659.30 | 2731.30 | 8891.97 |
| Credit to private sector (%GDP) | 14.8535 | 27.0366 | 46.4211 |
| Liquid Liabilities (%GDP) | 24.5790 | 56.1961 | 58.0585 |
| Deposit money bank assets (%GDP) | 20.9130 | 45.8504 | 56.8155 |
| Bank deposits (%GDP) | 17.6861 | 37.3495 | 50.3873 |

All the indicators are calculated as the average of 1981–2011; See Appendix A for list of selected Non-Oil Dependent Countries; Source: Data on GDP Per Capita is collected from World Development Indicators Database http://databank.worldbank.org/data/databases.aspx; while data on Financial Intermediary Indicators is collected from Beck, et al., (2000) Financial Development and Structure Dataset (2013 revised version) [18].

Table 1 illustrates the conclusion of a large body of cross-country, panel and time-series evidence: countries with high level of financial intermediary development achieved higher average annual GDP per capita over the period 1981–2011. With higher level of financial intermediary development, the non-oil dependent countries experienced higher average level of GDP per capita over the period. Between the two OPEC member countries (Nigeria and Algeria), Algeria achieved the highest level of GDP per capita over the period with higher level of financial intermediary development over Nigeria. The figures in Table 1 highlight the inability of the Nigerian financial intermediary sector to extend credit to the private sector. The average domestic credit by domestic banks and other financial intermediaries to the Nigerian private sector stood at approximately 14.55% over the period. Comparatively, the figure is by far lower than the average amount of credit provided by financial intermediaries in Algeria and the non-oil dependent countries over the period. Nigeria's average GDP per capita stood at US$659.30 over the period. This figure places Nigeria among the poorest countries in the distribution and illustrates the conclusion of many studies on natural resource-dependent economies: natural resource-dependent economies tend to grow slower than non-natural resource-dependent economies (Sachs and Warner, 1995 [19], 2001 [20]; Auty, 2001 [21]).

## 3. Data and Methodology

### 3.1. Data Description

This study uses annual data covering the period from 1981 to 2011. Economic growth is defined as the real GDP per capita. Four widely used measures of financial sector intermediary development: the domestic bank credit to the private sector divided by GDP, Liquid Liabilities to GDP, Deposit money bank assets to GDP and Bank deposits to GDP are employed to capture various aspects of the financial sector intermediary activities in Nigeria. Four control variables are included to capture other components of the Nigerian macroeconomic environment that could influence the growth of the Nigerian economy. The variables include: the international crude oil price measured as the Brent spot price (in US dollars per barrel), the ratio of total trade (exports plus imports) to GDP which explains the degree of openness of the Nigerian economy to trade, Gross fixed capital formation (% of GDP) and General government final consumption expenditure (% of GDP). Time series data on the variables are sourced from World Development Indicators database, World Bank (Online), Beck et al. (2000) Financial Development and Structure Dataset (2013 revised version) [18] and BP Statistical Review of World Energy. Appendix B provides data associated with these variables.

**Table 2.** Eigenvalues, Proportion and Eigenvectors of each first principal component.

|  | FDindex1 | FDindex2 | >FDindex3 |
|---|---|---|---|
| Eigenvalues | 3.6866 | 2.6983 | 2.7366 |
| Proportion | 0.9216 | 0.8994 | 0.9122 |
| **Eigenvectors (Loadings)** | | | |
| CPS | 0.4940 | 0.5799 | 0.5663 |
| LIQ | 0.4750 | 0.5522 | 0.5629 |
| BA | 0.5112 | 0.5990 |  |
| BD | 0.5186 |  | 0.6021 |

Note: CPS is for Credit to private sector (%GDP); LIQ is for Liquid Liabilities (%GDP); BA is for Deposit money bank assets (%GDP) and BD is for Bank deposits (%GDP).

The four selected indicators of financial development are used to construct three composite measures for financial sector intermediary development in Nigeria using principal component analysis that significantly addresses the possible high correlation between these indicators. Table 2 presents the eigenvalues, the proportion of the variance explained and the eigenvector of each first principal component from which three financial intermediary development indices are constructed.

The eigenvector values are taken as the weights of the variables in the construction of each of the three indices. FDindex1 is constructed as a linear combination of all the four financial intermediary development indicators (CPS, LIQ, BA and BD), capturing approximately 92% of the total variations in the four indicators. FDindex2 explains about 90% of the total variations in the linear combination of CPS, LIQ and BA. FDindex3 is defined as a linear combination of CPS, LIQ and BD explaining approximately 91% of the total variations in the three variables.

### 3.2. Unit Root Test

The order of integration of the variables is investigated first. The stationarity tests is performed first in levels and then in first difference to establish the presence of unit roots and the order of integration in all the variables. The results of the Augmented Dickey-Fuller (ADF) and Phillips-Perron (PP) stationarity tests in Table 3 show that the variables are integrated of different order; while *ln*Invest is stationary at level I(0) other variables are integrated of order one I(1).

**Table 3.** ADF and PP Unit root tests.

| | In Level I(0) | | First Difference I(1) | |
|---|---|---|---|---|
| **Variable** | **ADF** | **PP** | **ADF** | **PP** |
| *lnRGDPC* | 0.3916 | 0.0858 | −4.0543 *** | −4.0409 *** |
| *lnFDindex1* | −1.8464 | −1.3791 | −4.1177 *** | −3.9394 *** |
| *lnFDindex2* | −1.1789 | −1.4047 | −4.0985 *** | −3.9081 *** |
| *lnFDindex3* | −1.2914 | −1.5943 | −4.1943 *** | 4.0331 *** |
| *lnCPS* | −2.0362 | −1.4202 | −3.9147 *** | −3.6199 ** |
| *lnLIQ* | −2.0383 | −1.7897 | −4.3686 *** | −4.2594 *** |
| *lnBA* | −0.9044 | −0.9044 | −3.8804 *** | −3.7139 *** |
| *lnBD* | −1.0851 | −1.3217 | −4.1300 *** | −3.9828 *** |
| *lnOilp* | −0.0940 | −0.0011 | −5.5307 *** | −5.5316 *** |
| *lnOpen* | −2.0651 | −1.9280 | −7.3304 *** | 7.3284 *** |
| *lnInvest* | −2.7740 * | −2.7659 * | −4.9084 *** | −4.6078 *** |
| *lnGCExp* | −2.5606 | −2.5476 | −6.0260 *** | −6.0548 *** |

* Significance at 10%, ** Significance at 5%, *** Significance at 1%. The asterisks indicate the rejection of the null hypothesis of unit root. All the variables are in the natural log form.

### 3.3. Empirical Methodology

This study links economic growth financial development controlling for the influence of crude oil price, trade openness, rate of investment and government consumption expenditure. This relationship is given in the long-linear empirical model below:

$$lnRgdp = \alpha_0 + \alpha_1 lnFD + \alpha_2 lnOilp + \alpha_3 lnOpen + \alpha_4 lnInvest + \alpha_4 lnGCExpt + \varepsilon_t \qquad (1)$$

where *ln* denotes the natural logarithm function; $\varepsilon_t$ is the error term; *Rgdp* represents real GDP per capita for economic growth; *FD* represents financial development indicators (*lnFDindex1*, *lnFDindex2*, *lnFDindex3*, *lnCPS*, *lnLIQ*, *lnBA* and *lnBD*); *Oilp* is the international crude oil price; *Open* is for trade openness; *Invest* is for the rate of investment represented by the ratio of gross fixed capital formation to GDP and *GCExpt* is for government consumption expenditure.

Following Nwani and Bassey Orie (2016 [6]) and Nwani et al. (2016 [22]) this study employs the Autoregressive Distributed Lag (ARDL-Bounds) testing approach to co-integration proposed by Pesaran et al. (2001 [23]). The ARDL approach offers some desirable statistical advantages over other co-integration techniques. While other co-integration techniques require all the variables to be integrated of the same order, ARDL test procedure provides valid results whether the variables are I(0) or I(1) or mutually co-integrated and provides very efficient and consistent test results in small and large sample sizes (see Pesaran et al., 2001 [23]). The small number of observations and the

different order of integration make ARDL the preferred approach in this study. The ARDL model can be specified as:

$$
\begin{aligned}
\Delta lnGrowth_t = \quad & \beta_0 + \sum_{i=1}^{n} \beta_{1i}\Delta lnGrowth_{t-i} + \sum_{i=0}^{n} \beta_{2i}\Delta lnFD_{1t-i} + \sum_{i=0}^{n} \beta_{3i}\Delta lnOilp_{2_{t-i}} \\
& + \sum_{i=0}^{n} \beta_{4i}\Delta lnOpen_{t-i} + \sum_{i=0}^{n} \beta_{5i}\Delta lnInvest_{4_{t-i}} + \sum_{i=0}^{n} \beta_{6i}\Delta lnGCExp_{5_{t-i}} \\
& + \beta_7 lnGrowth_{t-1} + \beta_8 lnFD_{t-1} + \beta_9 lnOilP_{t-1} + \beta_{10} lnOpen_{t-1} \\
& + \beta_{11} lnInvest_{t-1} + \beta_{12} lnGCExp_{t-1} + \varepsilon_{1t}
\end{aligned}
\tag{2}
$$

where $\Delta$ is the difference operator. The test involves conducting *F*-test for joint significance of the coefficients of lagged variables for the purpose of examining the existence of a long-run relationship among the variables. The null hypothesis of no long-run relationship existing between the variables ($H_o$: $\beta_7 = \beta_8 = \beta_9 = \beta_{10} = \beta_{11} = \beta_{12} = 0$) is examined following Pesaran et al. (2001 [23]). The decision to reject or accept $H_o$ is based on the Following conditions: if *F*-value > upper bound, then reject $H_o$ and the variables are co-integrated, if *F*-value < lower bound, then accept $H_o$ and the variables are not co-integrated, but if *F*-value $\geq$ lower bound and $\leq$ upper bound, then the decision is inconclusive. The error correction model for the estimation of the short run relationships is specified as:

$$
\begin{aligned}
\Delta lnGrowth_t = \quad & \beta_0 + \sum_{i=1}^{n} \beta_{1i}\Delta lnGrowth_{t-i} + \sum_{i=0}^{n} \beta_{2i}\Delta lnFD_{1t-i} + \sum_{i=0}^{n} \beta_{3i}\Delta lnOilp_{2_{t-i}} \\
& + \sum_{i=0}^{n} \beta_{4i}\Delta lnOpen_{t-i} + \sum_{i=0}^{n} \beta_{5i}\Delta lnInvest_{4_{t-i}} + \sum_{i=0}^{n} \beta_{6i}\Delta lnGCExp_{5_{t-i}} \\
& + \lambda_1 ECM_{t-1} + u_{1t}
\end{aligned}
\tag{3}
$$

A negative and significant $ECM_{t-1}$ coefficient ($\lambda_1$) implies that any short term disequilibrium between the dependent and explanatory variables will converge back to the long-run equilibrium relationship. This study implements the ARDL bounds cointegration analysis using Eviews 9.5.

## 4. Empirical Results

### 4.1. Co-Integration Analysis

Using four indicators of financial development and three financial development indices constructed from the four indicators using principal component analysis (PCA), this study tested for co-integration on seven alternative specifications employing one measure of financial intermediary development at a time. The results of the co-integration test based on the ARDL-bounds testing method, implemented using Eviews 9.5 software are presented in Table 4. The results indicate that in all the specifications, the *F*-statistic is greater than the upper critical bound at 5% significance level. This study therefore rejects the null hypothesis of no co-integration. This shows that there is a long-run causal relationship among the variables in all the specifications.

**Table 4.** Results of Cointegration test.

| Models | | | *F*-Statistic | Result |
|---|---|---|---|---|
| 1 | $F_{RGDPC}$(RGDPC | FDindex1, Oilp, Open, Invest, GCExp) | ARDL(1, 1, 1, 1, 0, 0) | 4.8722 *** | Cointegration |
| 2 | $F_{RGDPC}$(RGDPC | FDindex2, Oilp, Open, Invest, GCExp) | ARDL(1, 1, 1, 1, 0, 0) | 4.8089 *** | Cointegration |
| 3 | $F_{RGDPC}$(RGDPC | FDindex3, Oilp, Open, Invest, GCExp) | ARDL(1, 1, 1, 1, 0, 0) | 4.8305 *** | Cointegration |
| 4 | $F_{RGDPC}$(RGDPC | CPS, Oilp, Open, Invest, GCExp) | ARDL(1, 1, 1, 1, 0, 0) | 4.5053 *** | Cointegration |
| 5 | $F_{RGDPC}$(RGDPC | LIQ, Oilp, Open, Invest, GCExp) | ARDL(1, 0, 1, 1, 0, 0) | 4.0349 ** | Cointegration |
| 6 | $F_{RGDPC}$(RGDPC | BA, Oilp, Open, Invest, GCExp) | ARDL(1, 1, 1, 1, 0, 0) | 4.8738 *** | Cointegration |
| 7 | $F_{RGDPC}$(RGDPC | BD, Oilp, Open, Invest, GCExp) | ARDL(1, 1, 1, 1, 0, 0) | 5.0236 *** | Cointegration |
| | Critical Value Bounds | 1% | 5% | 10% |
| | I0 Bound | 3.06 | 2.39 | 2.08 |
| | I1 Bound | 4.15 | 3.38 | 3.0 |

Notes: ARDL Models selected on Akaike info criterion (AIC), Restricted intercept and no trend for *k* = 5; ** and *** indicate significance at 5% and 1%, respectively; Source of Asymptotic critical value bounds: Pesaran et al. (2001) [23].

*4.2. Long-Run and Short-Run Estimates*

　　The estimated long-run coefficients of the seven ARDL specifications are presented in Table 5. In all the specifications, the coefficient of financial intermediary development is negative and insignificant. This indicates that financial intermediary development has a negative but insignificant long-run effect on economic growth, suggesting that financial development does not stimulate economic growth in Nigeria. The empirical results show that in all the specifications, the coefficient of oil price is positive and highly significant, indicating that oil price is the long-run driver of economic growth in Nigeria. Trade openness, government consumption expenditure and gross fixed capital formation are all found to be insignificant. The long-run coefficients in Table 5 create a picture of an economy significantly dominated by activities in the oil sector. A 1% increase in oil price increases the growth of the economy by over 0.40% in the long-run. Given that oil price is significantly determined in the international oil market and not by domestic activities and performance (Quixina and Almeida, 2014 [12]; Samargandi et al., 2014 [13]), a fall in crude oil price would adversely affect the performance of the economy. Specifically, a 1% decrease in crude oil price would cause the level of economic growth to decline by over 0.40%.

**Table 5.** Long-run Coefficients.

| Variables | 1 | 2 | 3 | 4 | 5 | 6 | 7 |
|---|---|---|---|---|---|---|---|
| C | 5.4004 *** [11.0828] | 5.3955 *** [11.1151] | 5.4014 *** [11.1847] | 5.2120 *** [12.3461] | 5.7642 *** [12.1388] | 5.3265 *** [12.0457] | 5.3343 *** [12.3868] |
| *ln*Fintdex | −0.0738 [−0.7921] | | | | | | |
| *ln*Fintdex2 | | −0.0741 [−0.7834] | | | | | |
| *ln*Fintdex3 | | | −0.0794 [−0.8083] | | | | |
| *ln*CPS | | | | −0.05191 [−0.5554] | | | |
| *ln*LIQ | | | | | −0.1388 [−0.3635] | | |
| *ln*BA | | | | | | −0.0565 [−0.7240] | |
| *ln*BD | | | | | | | −0.0705 [−0.8067] |
| *ln*Oilp | 0.4447 *** [7.8444] | 0.4439 *** [7.7717] | 0.4433 *** [8.0862] | 0.4398 *** [6.5777] | 0.4292 *** [9.6603] | 0.4426 *** [7.5041] | 0.4457 *** [8.0845] |
| *ln*Open | 0.0183 [0.2455] | 0.0179 [0.2376] | 0.0205 [0.2778] | 0.0446 [0.5883] | −0.0311 [−0.4476] | 0.0145 [0.1883] | 0.0195 [0.2657] |
| *ln*Invest | −0.0402 [−0.7028] | −0.0419 [−0.7312] | −0.0347 [−0.5889] | −0.0552 [−0.8967] | −0.0050 [−0.0839] | −0.0521 [−0.9258] | −0.0355 [−0.6163] |
| *ln*GCExp | −0.0552 [−0.9708] | −0.0523 [−0.9180] | −0.0602 [−1.0462] | −0.0506 [−0.8251] | −0.0701 [−1.2131] | −0.0431 [−0.7635] | −0.0646 [−1.1287] |

Note: *** indicate significance at 1%, *t*-statistics in [].

　　The coefficients of the error correction model for all the seven specifications are given in Table 6. The coefficient of ECT (−1) are all negative and significant at 1% level, suggesting that short-run disequilibrium is corrected in the long-run equilibrium. The short-run coefficient of financial intermediary development in all the seven specifications is negative and significant at 5% level. The significance of the short-run negative effect of financial intermediary development on economic growth in Nigeria highlights the high degree of inefficiency in resource mobilization and allocation in the Nigerian financial intermediary sector. The short-run coefficient of crude oil price is positive and

significant at 1% level. The results confirm the dominant role of the oil sector on economic activities in Nigeria. A 1% increase in the oil price results in more than 0.12% increase in the level of economic growth in the short-run. The positive relationship between oil price and economic growth in Nigeria also indicate that a 1% decrease in oil price will result in more than 0.12% decrease in the level of economic growth in Nigeria in the short-run. Surprisingly, the coefficient of trade openness is found to be negative and significant at 1% level. The rate of investment and size of the public sector are all found to be insignificant.

**Table 6.** Short-run error correction estimates.

| Variables | 1 | 2 | 3 | 4 | 5 | 6 | 7 |
|---|---|---|---|---|---|---|---|
| ECM ($-1$) | $-0.5275$ *** [$-5.4830$] | $-0.5263$ *** [$-5.4450$] | $-0.5264$ *** [$-5.4833$] | $-0.4990$ *** [$-5.2639$] | $-0.5490$ *** [$-5.4517$] | $-0.5276$ *** [$-5.4231$] | $-0.5303$ *** [$-5.5753$] |
| $\Delta ln$Fintdex1 | $-0.1325$ ** [$-2.5617$] | | | | | | |
| $\Delta ln$Fintdex2 | | $-0.1315$ ** [$-2.5102$] | | | | | |
| $\Delta ln$Fintdex3 | | | $-0.1344$ ** [$-2.5432$] | | | | |
| $\Delta$lnCPS | | | | $-0.1102$ ** [$-2.2139$] | | | |
| $\Delta$lnLIQ | | | | | $-0.1372$ ** [$-2.5241$] | | |
| $\Delta$lnBA | | | | | | $-0.1182$ ** [$-2.5249$] | |
| $\Delta ln$BD | | | | | | | $-0.1298$ ** [$-2.6696$] |
| $\Delta ln$Oilp | 0.1270 *** [3.9324] | 0.1272 *** [3.9161] | 0.1227 *** [3.7626] | 0.1195 *** [3.5139] | 0.1287 *** [3.9577] | 0.1384 *** [4.3448] | 0.1275 *** [4.0102] |
| $\Delta ln$Open | $-0.1114$ *** [$-3.2854$] | $-0.1117$ *** [$-3.2776$] | $-0.1083$ *** [$-3.2150$] | $-0.0999$ *** [$-2.9220$] | $-0.1241$ *** [$-3.6337$] | $-0.1178$ *** [$-3.4000$] | $-0.1101$ *** [$-3.2927$] |
| $\Delta ln$Invest | 0.0055 [0.1435] | 0.0047 [0.1223] | 0.0078 [0.2035] | $-0.0007$ [$-0.0166$] | 0.0255 [0.6782] | 0.0002 [0.0056] | 0.0079 [0.2078] |
| $\Delta ln$GCexp | $-0.0012$ [$-0.0403$] | 0.0001 [0.0029] | $-0.0041$ [$-0.1375$] | 0.0001 [0.0035] | $-0.0099$ [$-0.3235$] | 0.0060 [0.1995] | $-0.0057$ [$-0.1934$] |
| *Diagnostic tests* | | | | | | | |
| Adj. $R^2$ | 0.9542 | 0.9539 | 0.9542 | 0.9523 | 0.9546 | 0.9539 | 0.9550 |
| D-W stat | 1.8343 | 1.8357 | 1.8181 | 1.8135 | 1.7761 | 1.8645 | 1.8259 |
| SC | 0.2413 (0.5396) | 0.2416 (0.5394) | 0.2869 (0.5041) | 0.3399 (0.4677) | 0.4415 (0.4209) | 0.1731 (0.6028) | 0.2528 (0.5303) |
| Het | 1.0837 (0.2901) | 1.0667 (0.2938) | 1.0877 (0.2893) | 0.6503 (0.4089) | 1.2312 (0.2607) | 0.9728 (0.3153) | 1.1130 (0.2839) |

Notes: Adj. $R^2$ means Adjusted $R$-squared. SC is the Breusch-Godfrey serial correlation LM test. Het is the ARCH test for heteroscedasticity ** and *** indicate significance at 5% and 1%, respectively. *t*-statistics in [], *p*-values in ().

Overall, the results of this study show that the effect of financial development on economic growth in Nigeria is not different from what has been reported from other OPEC member countries. Specifically, the results are in line with what Cevik and Rahmati (2013 [11]), Quixina and Almeida (2014 [12]) and Samargandi et al. (2014 [13]) observed for Libya, Angola and Saudi Arabia respectively, confirming the insignificant effect of financial intermediary development on economic growth in oil-dependent economies as documented by Nili and Rastad (2007 [15]), Beck (2011 [16]) and Barajas, et al. (2013 [17]). The results therefore highlight the specific feature of oil-exporting economies. Economic activities are driven by oil price determined in the international oil market and not by domestic economic

activities (Samargandi et al., 2014 [13]); creating economic conditions in oil-dependent economies that adversely affect the ability of financial intermediaries to allocate resources efficiently. Nili and Rastad (2007 [15]) identified these economic conditions to include the general inefficiency of financial institutions, the dominant role of public sector in resource allocation and the weakness of private sector, while Beck (2011 [16]) shows that rent-seeking associated with the high revenue from the oil sector drives resources away from institutions that could encourage economic activities in the private sector to the inefficient public sector.

*4.3. Diagnostic and Stability Tests*

From the diagnostic test results (see results in Table 6), there is no evidence of serial correlation and heteroscedasticity in each of the Autoregressive Distributed Lag (ARDL-Bounds) models specified. Figures 1–7 show that the cumulative sum of recursive residuals (CUSUM) and the Cumulative Sum of Squares of Recursive Residuals (CUSUMSQ) are within the critical boundaries for the 5% significance level indicating that the coefficients of the ARDL model in each of the specifications are stable.

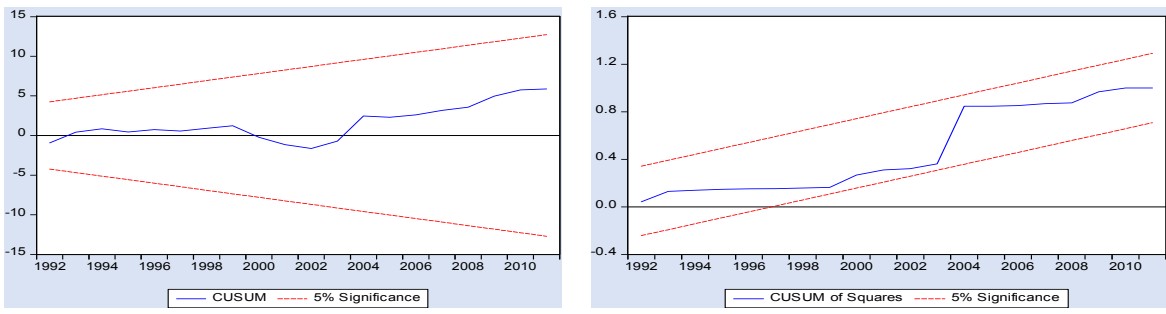

**Figure 1.** Plot of CUSUM and CUSUMQ for coefficient stability of Error Correction term (ECM ) model 1.

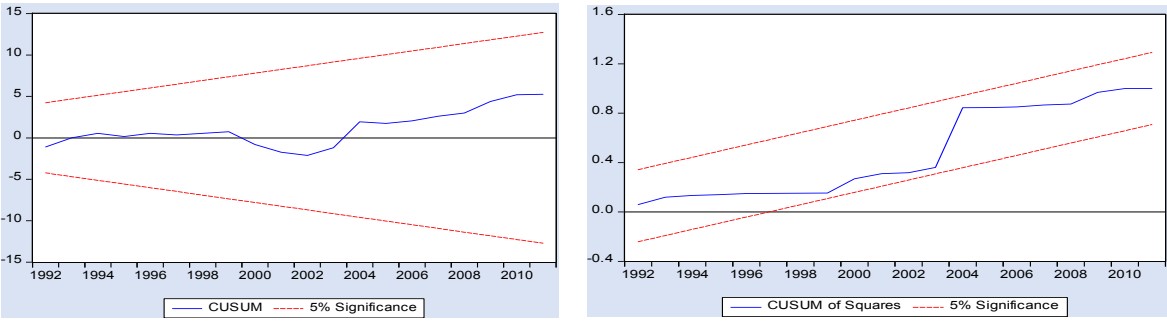

**Figure 2.** Plot of CUSUM and CUSUMQ for coefficient stability of ECM model 2.

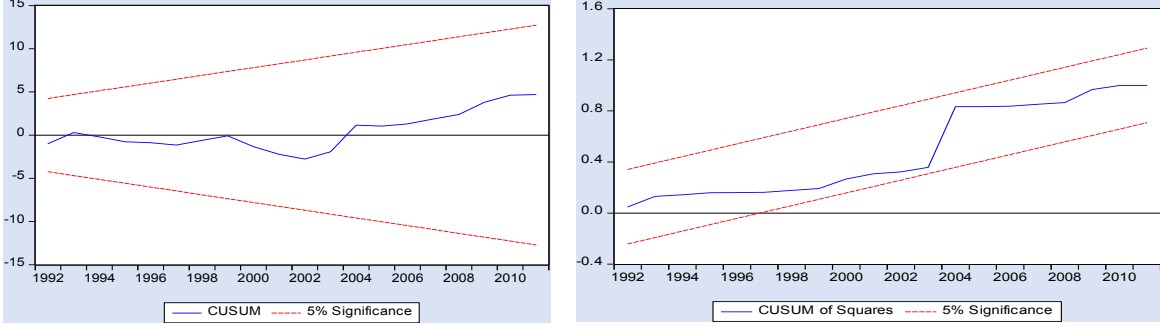

**Figure 3.** Plot of CUSUM and CUSUMQ for coefficient stability of ECM model 3.

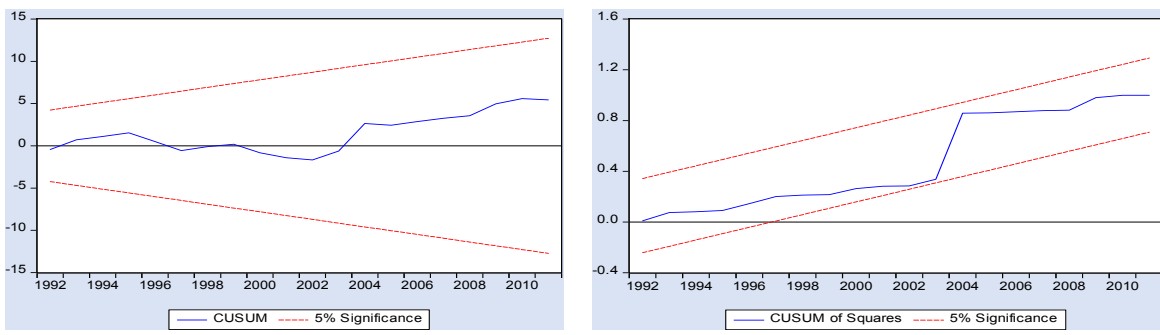

**Figure 4.** Plot of CUSUM and CUSUMQ for coefficient stability of ECM model 4.

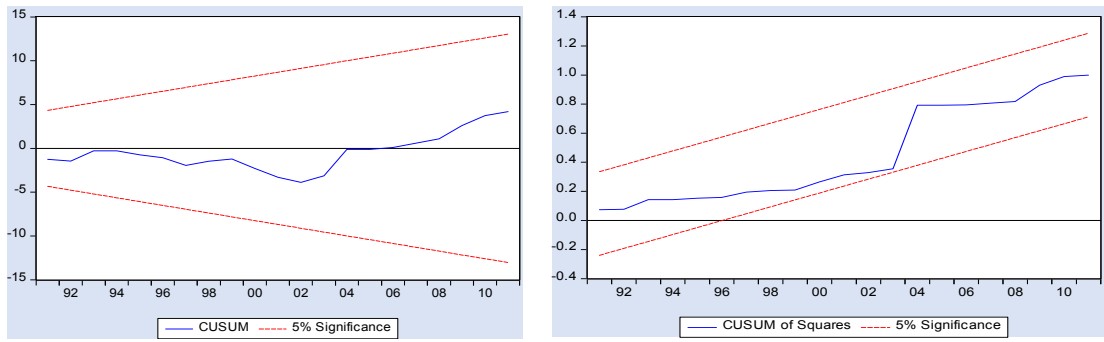

**Figure 5.** Plot of CUSUM and CUSUMQ for coefficient stability of ECM model 5.

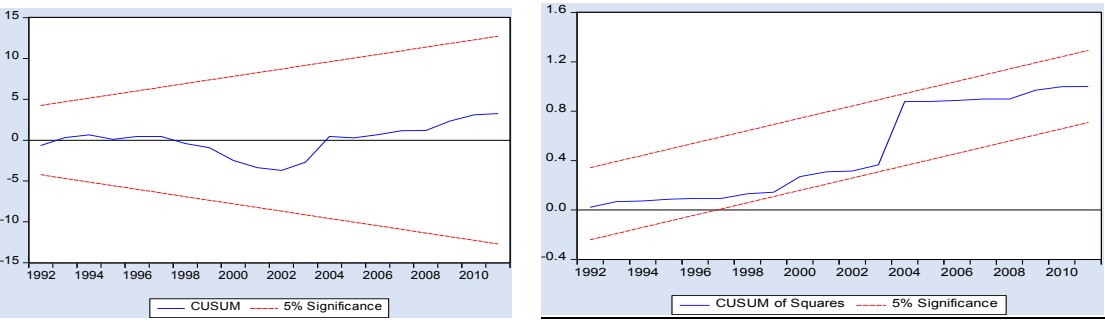

**Figure 6.** Plot of CUSUM and CUSUMQ for coefficient stability of ECM model 6.

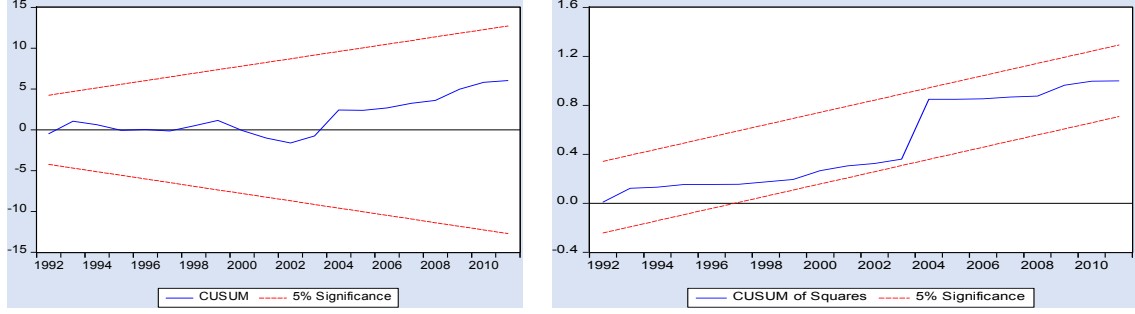

**Figure 7.** Plot of CUSUM and CUSUMQ for coefficient stability of ECM model 7.

## 5. Concluding Remarks

Motivated by the growing interest among researchers and policy makers in understanding the effects of financial sector intermediary development on economic growth and the limited attention

that has been given to the special case of oil-dependent economies, this study empirically examined the effect of financial development on economic growth in Nigeria over the period 1981 to 2011 using the auto-regressive distributed lag (ARDL) approach to co-integration analysis, controlling for the possible effects of oil price, trade openness, gross fixed capital formation and government consumption expenditure on economic activities in Nigeria. The results indicate that financial development in Nigeria has insignificant negative effect on economic growth in the long-run and significant negative effect in the short-run. The case of Nigeria is therefore not different from what has been observed generally in oil-dependent economies as documented in Nili and Rastad (2007 [15]), Beck (2011 [16]) and Barajas et al. (2013 [17]). This economic condition might be associated with the dominant role of the public sector and the weakness of the private sector in generating economic activities in the economy as a result the dependence of the economy on oil wealth. Interestingly, the influence of crude oil price on economic growth in the economy is found positive and significant in the long-run and short-run.

The dominant role of oil price on economic activities calls for urgent need to diversify the Nigerian economy. Diversified economies are less exposed to external and fiscal shocks. The development of the financial system could be the right strategy to achieving this objective giving the link between the financial sector and the private sector. Strengthening the intermediary role of financial intermediaries in Nigerian economy by building institutional framework that would channel financial resources in the economy to productive investment projects through the financial system could stimulate economic activities in the private sector, lessen the burden on the public sector and the dominance of the oil sector in the economy.

**Acknowledgments:** The valuable comments and suggestions of the anonymous Reviewers and the Editor are highly appreciated.

**Conflicts of Interest:** The author declares no conflict of interest.

## Appendix A. List of Selected Non-Oil Dependent Countries

| List of Selected Non-Oil Dependent Countries | | |
|---|---|---|
| **Country Name** | **Country Name** | **Country Name** |
| Argentina | India | Papua New Guinea |
| Australia | Spain | Paraguay |
| Bahamas, The | Ireland | Peru |
| Botswana | Italy | Portugal |
| Burkina Faso | Jamaica | Senegal |
| Cote d'Ivoire | Japan | Seychelles |
| Cyprus | Kenya | Singapore |
| Dominica | Korea, Rep. | Sri Lanka |
| Dominican Republic | Madagascar | St. Lucia |
| Fiji | Malawi | Swaziland |
| Finland | Malaysia | Thailand |
| Gambia, The | Malta | Turkey |
| Greece | Mauritius | United States |
| Grenada | Nepal | Uruguay |
| Guatemala | Pakistan | Vanuatu |
| Honduras | Panama | |

## Appendix B. Data Associated with the Study

| Year | Liquid Liabilities to GDP (%) | Deposit Money Bank Assets to GDP (%) | Private Credit By Deposit Money Banks to GDP (%) | Bank Deposits To GDP (%) | GDP per Capita (Constant 2005 US$) | Gross Fixed Capital Formation (% of GDP) | (% of GDP) | Trade (% of GDP) | Crude Oil Price (Brent UD$ per Barrel) |
|------|------|------|------|------|------|------|------|------|------|
| 1981 | 30.92 | 20.05 | 14.27 | 20.30 | 710.6061 | 35.22126 | 14.66089 | 48.29332 | 35.93 |
| 1982 | 31.63 | 22.55 | 16.79 | 20.93 | 685.0285 | 31.95333 | 15.66617 | 37.7485 | 32.97 |
| 1983 | 33.10 | 25.50 | 17.24 | 22.12 | 634.1195 | 23.0065 | 15.33589 | 27.03717 | 29.55 |
| 1984 | 36.17 | 30.15 | 17.11 | 24.19 | 605.7565 | 14.22397 | 13.1504 | 23.60888 | 28.78 |
| 1985 | 33.62 | 30.00 | 15.27 | 22.26 | 639.5429 | 11.96524 | 12.73176 | 25.90006 | 27.56 |
| 1986 | 34.98 | 29.73 | 17.94 | 23.11 | 568.5368 | 15.15382 | 12.58138 | 23.71676 | 14.43 |
| 1987 | 26.13 | 20.60 | 14.15 | 17.68 | 494.239 | 13.60753 | 7.20595 | 41.64666 | 18.44 |
| 1988 | 26.76 | 19.62 | 13.08 | 18.05 | 517.6942 | 11.87108 | 7.645588 | 35.31198 | 14.92 |
| 1989 | 22.29 | 13.87 | 10.29 | 13.89 | 536.9417 | 11.74232 | 5.446973 | 60.39176 | 18.23 |
| 1990 | 20.76 | 11.59 | 8.78 | 12.45 | 590.0519 | 14.25014 | 4.964438 | 53.03022 | 23.73 |
| 1991 | 22.39 | 11.24 | 8.38 | 13.68 | 571.6511 | 13.73268 | 4.833249 | 64.8766 | 20.00 |
| 1992 | 21.06 | 11.12 | 9.49 | 13.01 | 559.8226 | 12.74817 | 5.961901 | 61.03097 | 19.32 |
| 1993 | 25.80 | 16.05 | 12.71 | 16.54 | 557.3815 | 13.55003 | 6.542752 | 58.10985 | 16.97 |
| 1994 | 25.88 | 17.48 | 12.37 | 16.43 | 548.5813 | 11.16543 | 17.94384 | 42.30887 | 15.82 |
| 1995 | 15.98 | 11.41 | 9.16 | 9.81 | 533.4169 | 7.065756 | 12.08512 | 59.76783 | 17.02 |
| 1996 | 12.86 | 9.98 | 8.41 | 8.12 | 546.2431 | 7.289924 | 10.01718 | 57.69099 | 20.67 |
| 1997 | 13.87 | 11.60 | 9.71 | 9.10 | 547.6899 | 8.356764 | 12.99717 | 76.85999 | 19.09 |
| 1998 | 16.71 | 13.71 | 11.84 | 11.12 | 548.6618 | 8.60161 | 13.97338 | 66.17325 | 12.72 |
| 1999 | 18.80 | 16.52 | 12.47 | 13.16 | 537.6261 | 6.994108 | 6.982946 | 55.84639 | 17.97 |
| 2000 | 17.53 | 15.52 | 10.42 | 12.65 | 552.1869 | 7.017881 | 8.34258 | 71.38053 | 28.50 |
| 2001 | 25.08 | 20.73 | 14.82 | 18.16 | 562.2306 | 7.579868 | 8.210655 | 81.81285 | 24.44 |
| 2002 | 20.70 | 17.77 | 12.58 | 15.32 | 568.9709 | 7.009923 | 6.709872 | 63.38364 | 25.02 |
| 2003 | 18.98 | 17.26 | 12.12 | 14.37 | 612.1304 | 9.904054 | 5.152789 | 75.2189 | 28.83 |
| 2004 | 16.39 | 15.83 | 11.51 | 12.72 | 797.8757 | 7.39337 | 6.731593 | 48.44813 | 38.27 |
| 2005 | 16.81 | 17.01 | 12.32 | 13.16 | 804.1524 | 5.458996 | 6.807476 | 50.74836 | 54.52 |
| 2006 | 16.92 | 17.27 | 12.10 | 13.41 | 847.5391 | 8.265865 | 6.859525 | 64.60931 | 65.14 |
| 2007 | 22.54 | 26.61 | 18.16 | 19.06 | 881.5914 | 9.249637 | 10.18013 | 64.46291 | 72.39 |
| 2008 | 30.12 | 36.15 | 27.37 | 26.78 | 911.9575 | 8.323477 | 11.64139 | 64.97297 | 97.26 |
| 2009 | 37.70 | 43.53 | 35.39 | 34.06 | 949.0064 | 12.08816 | 12.95737 | 61.80285 | 61.67 |
| 2010 | 36.49 | 41.55 | 31.29 | 32.95 | 995.6802 | 16.99081 | 8.711384 | 42.65138 | 79.50 |
| 2011 | 32.99 | 36.34 | 22.91 | 29.68 | 1015.815 | 15.97916 | 8.494303 | 52.7941 | 111.26 |

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
