# Peer review of "The Impact of Financial Development on Economic Growth in Nigeria: An ARDL Analysis"

_economies, doi:10.3390/economies4040026_

Reviewer 1 Report

First, I found a better paper, already published, similar:

1. The author should describe the software used in a note.

2. Include in the end the data used.

3. In the case of Nigeria, it may go causality from economic growth to financial development and not viceversa.

4. The dynamic development of Nigeria is different from that observed by countries from other continents. Avoid applying by applying a technique, eliminating the historical context.

5. What explanations offered regarding the negative relationship between financial development and economic growth? Your answer should be based on the experience of the authors. 

6. Do they change the results considering the series of financial development without dividing between the GDP?

7. In defining the variables used, put from the beginning the identifiers listed in the tables.

Author Response

Responses for Reviewer 1:

The valuable comments and suggestions of Reviewer 1 are highly appreciated. These changes recommended by the reviewers have improved the quality of the study. 

1.      The author should describe the software used in a note.

Response: The software used is Eviews 9.5 – this has been included as part of the description of the methodology adopted in the study. Thank you.

2.      Include in the end the data used.

Response: The data used in the study, sources and definition of variables are all in the supplementary file attached to the study. Thank you.

3.      In the case of Nigeria, it may go causality from economic growth to financial development and not vice versa.

Response: This study however focuses only on examining the effects of financial development on economic growth in Nigeria. Thank you.

4.      The dynamic development of Nigeria is different from that observed by countries from other continents. Avoid applying by applying a technique, eliminating the historical context.

Response: Purely empirical study with crude oil price included to capture the role of oil wealth in the Nigerian economy. Thank you.

5.      What explanations offered regarding the negative relationship between financial development and economic growth? Your answer should be based on the experience of the authors. 

Response: More explanation on the results and implications has been included in the concluding remarks to the study. Thank you.

6.      Do they change the results considering the series of financial development without dividing between the GDP?

Response:  No. Thank you

7.      In defining the variables used, put from the beginning the identifiers listed in the tables.

      Response: All the variables used in the study are defined as recommended. Thank you

Reviewer 2 Report

The authors of this paper used the auto-regressive distributed lag (ARDL) method to investigate the relationship between financial development and growth in Nigeria during the period: 1981-2011. They found that this relationship is negative and insignificant in the long-run but negative and significant in the short-tun.

The paper can be improved in several aspects:

(i) The authors need to clarify in the introduction how the ultimate question is related to the existing literature, and what is their contribution using Nigeria as a case study.

(ii) As they mentioned in their introduction, this research has been investigated by Barajas et al. (2013) (IMF Working Papers) using a dataset for 150 countries over the period 1975-2005 and dynamic panel estimation. 

(iii) The contribution and the relevance of the empirical strategy used must be properly assessed and compare to the paper of Barajas et al. (2013).

(iv) What is the economic rationale for just comparing Nigeria and Algeria? Why not Angola, for example?

(v) I suggest to the authors to control for some institutional variables (ease of doing business, property rights, corruption) and human capital as well.

(vi) Give some intuitions about the obtained results. Why the relationship is negative and insignificant in the long-run but significant in the short-run? Clear explanation is needed.

(vi) References must be cited in alphabetical order.

Author Response

Responses for Reviewer 2:

The valuable comments and suggestions of Reviewer 2 are highly appreciated.

(i)                 How is the ultimate question related to the existing literature, and what is the contribution using Nigeria as a case study?

Response: I have added some clarification in the introduction on the objective and contribution of the study. Thank you.

(ii)               As they mentioned in their introduction, this research has been investigated by Barajas et al. (2013) (IMF Working Papers) using a dataset for 150 countries over the period 1975-2005 and dynamic panel estimation. 

Response: This study contributes to existing studies using the case of Nigeria, which may or may not be different from the result documented by Barajas et al. (2013) from a cross-section of 150 countries. Thank you.

(iii)             The contribution and the relevance of the empirical strategy used must be properly assessed and compare to the paper of Barajas et al. (2013). 

Response: Barajas et al. (2013) implemented a cross-sectional analysis of 150 countries, this present study implements a time series analysis of the case of Nigeria only. The objectives, contributions and relevance of the empirical strategy are well explained in the study. More explanations have also been included to enable easy understanding. Thank you.

(iv)             What is the economic rationale for just comparing Nigeria and Algeria? Why not Angola, for example?

Response: The comparison is just for illustration. I also compared to a collection of non-oil countries that does not include all the countries in the world. Angola, will offer the same explanation to the point I raised in that illustration. The empirical analysis focuses on Nigeria alone.

(v)               I suggest to the authors to control for some institutional variables (ease of doing business, property rights, corruption) and human capital as well.

       Response: This study focuses on the impact of indicators of financial development on energy consumption. I captured the role oil in the economy which is argued to be reason for other issues including ease of doing business, property rights, corruption) Thank you

(vi)             References must be cited in alphabetical order.

Response: I followed the referencing style I found from the journal website. Thank you.

Round  2

Reviewer 1 Report

Accept because the authors made changes in the document. 

Remember that:

I found a better job done, already published, similar

Author Response

Although the study the reviewer highlighted {http://www.wami-imao.org/sites/default/files/journals/v12n2_unit4.pdf} examined the relationship between financial development and economic growth in Nigeria, the study did not control for the influence of oil wealth on economic activities in the oil dependent economy. My study of the relationship between financial development and economic growth in Nigeria controls for the influence of oil on economic activities in Nigeria using crude oil price (see Samargandi et al. ( 2014) and Quixina and Almeida, (2014) for the case of Saudi Arabia and Angola respectively).

In addition, my study of the relationship between financial development and economic growth in Nigeria uses both individual indicators of financial development and composite proxies constructed using principal component analysis (PCA). The study the reviewer highlighted {http://www.wami-imao.org/sites/default/files/journals/v12n2_unit4.pdf} only employed individual indicators of financial development. The inclusion of composite measures of the level of financial development in my study is considered necessary given that no consensus evidence exists on which individual proxy most appropriate for measuring the depth of financial intermediation.

These points have been highlighted in my study among the major contributions of the study to existing studies. Thank you.

Reference:

Quixina, Y.; Almeida, A. Financial Development and Economic Growth in a Natural Resource Based Economy: Evidence from Angola. FEP Working Papers 2014, No: 542. Retrieved from: http://www.fep.upNaN/investigacao/workingpapers/wp542.pdf.

Samargandi, N.; Fidrmuc, J.; Ghosh, S. Financial development and economic growth in an oil-rich economy: The case of Saudi Arabia. Economic Modelling 2014 43, 267–278. doi: 10.1016/j.econmod.2014.07.042.

Reviewer 2 Report

The paper can now be published in Economies.